# UBERT: UNSUPERVISED ADAPTIVE EARLY EXITS IN BERT

## ABSTRACT

Inference latency is an issue in pre-trained networks like BERT due to their large size. To overcome this, side branches are attached at the intermediary layers with provision for early inference instead of inference only at the last layer. This facilitates the early exit of 'easy' samples and requires only 'hard' samples to pass through all layers, thus reducing inference latency. However, the hardness of the samples is unknown a priori. This leads to the question of how to exit so that the accuracy and latency are well balanced. Also, the optimal choice of parameters involved in deciding exits can depend on the sample domain and hence need to be adapted. We develop an online learning algorithm named UBERT to decide if a sample can exit early. The decisions are based on confidence in inference exceeding a threshold at each exit point, and the algorithm simultaneously learns the optimal thresholds for all the exits. UBERT learns the optimal threshold for the sample domain using confidence observed at the intermediary layers without requiring any ground truth labels. We perform extensive experiments on five datasets with one and two early exits. We compare the performance against the case with no early exits, i.e., all samples exit at the last layer. UBERT achieves a 10%-53% reduction in time with a drop in accuracy in the range of 0.3% - 5.7% with one early exit. For the case with two exits, the time reduction increases to 32%-70% with only a marginal drop in accuracy of 0.1%-3.9%. The anonymized source code is available at `https://anonymous.4open.science/r/UBERT-F2DF/README.md`.

## 1 INTRODUCTION

Inference time is critical in many real-time Natural Language Processing (NLP) applications. Large-scale pre-trained Neural Nets (NNs) like ELMo Peters et al. (1802), BERT Devlin et al. (2018), ALBERT Lan et al. (2019) GPT Radford et al. (2019), XLNet Yang et al. (2019) and RoBERTa Liu et al. (2019) offer high accuracy but suffer from inference latencies due to their large size. This makes it challenging to deploy them on mobile or other edge devices which are resource-constrained.

Many variants of the BERT, like DeeBERT Xin et al. (2020b) and ElasticBERT Liu et al. (2021b), facilitate inference at the intermediary layers of NNs through early exits. In this configuration, each sample must ascertain whether the inference can be completed at intermediary layers or at the last layer. The decisions of early exit are based on the confidence at the intermediary layers being above a threshold. Even though it is anticipated that the final layer of the NN can have better accuracy than the intermediate layer, the cost will be high. In the following, we consider the cost quantified in the form of inference time (latency). However, depending on the application, the cost can also be present as other factors like power and computational resources.

The threshold used to compare the confidence levels significantly impacts the amount of latency and accuracy: with a lower threshold, more samples exit early, but with a lower confidence value, leading to lower accuracy and lower latency. With a higher threshold, fewer samples exit early, leading to higher latency but improved accuracy. Hence, one has to set the threshold that optimally trade-off between latency and accuracy.

The threshold is often determined using a labeled dataset during training and serves as a crucial reference point for decision-making during inference. However, a significant challenge arises when deploying pre-trained models that are later deployed on samples whose latent distribution can be different from the training samples. This scenario, often encountered in real-world applications,

raises the following question: How can we effectively adapt the threshold of deployed pre-trained models when the latent distribution of incoming samples is unknown, and ground truth labels are unavailable?

The optimal threshold value depends on the distribution of confidence levels at the attached exit, which can vary depending on the data distribution. During inference, since data arrives in sequential order (online fashion), and ground-truth labels are unavailable, the problem is that of learning the optimal threshold in an online and unsupervised setting. We propose an online learning algorithm using the Multi-Armed Bandit Auer et al. (2002b) framework to address this problem.

Our algorithm, named Unsupervised Adaptive Early Exits in BERT (UBERT), learns a threshold from a set of thresholds that achieves optimal tradeoff between accuracy and latency. We extensively evaluate the performance of UBERT on five datasets viz. IMDB, MRPC, SciTail, SNLI, Yelp, and QQP to cover different types of classification tasks –sentiment, entailment, and question answering. In our evaluation of the algorithm, ElasticBERT underwent an initial pre-training phase on a specific dataset (pre-deployment phase). Subsequently, we assessed its performance by testing it on a distinct dataset that shares a similar task type but exhibits a variation in latent data distribution

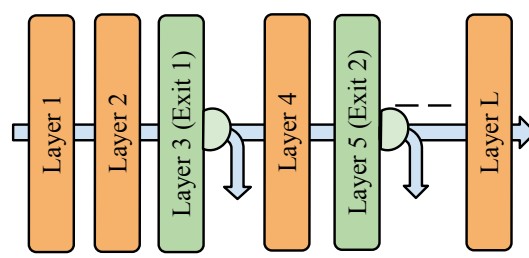

Figure 1: Early Exit setup: NNs are attached with exit layers so that 'easy' samples could be inferred without being processed till the final layer.

(post-deployment phase). For instance, we pre-trained ElasticBERT on the SST-2 dataset, which is primarily a sentiment classification task. Then, to gauge the adaptability of UBERT, we conducted evaluations on the IMDb and Yelp datasets, both of which involve review classification tasks akin to SST-2 but with notable differences in data distribution characteristics.

UBERT achieves substantial gains in inference speed, with reductions in inference time ranging from 10% to 53% in single-exit scenarios and from 32% to 72% in the two-exit cases, all while maintaining minimal loss in accuracy. UBERT decides the threshold on the fly based on the observations made on the previous samples. This stands in contrast to models like DeeBERT and ElasticBERT, which rely on fixed thresholds and offer limited insight into the rationale behind their threshold choices. Furthermore, these models require some degree of labeled data to fine-tune their threshold values.

It is important to highlight that the optimal threshold values can exhibit substantial variation when applied to different applications, as exemplified in Table 3. Utilizing a fixed threshold consistently across multiple layers and domains in an unsupervised context, especially when confronted with different latent data distributions, may yield suboptimal results. Consequently, UBERT sets itself apart by its capacity to dynamically acquire the optimal threshold values based on confidence metrics at each layer. This approach ensures peak performance across diverse scenarios and applications.

Our primary contributions can be summarized as follows: 1) We conceptualize the challenge of determining the optimal threshold for early exits within neural networks as an unsupervised online learning problem. 2) In Sec 4, we introduce an upper confidence-based algorithm named UBERT for the case with a single exit and discuss how it uses only the confidence to identify the optimal threshold and does not require any ground truth labels. 3) In Sec 5, we consider UBERT with multiple exits necessitating a distinct Multi-Armed Bandit (MAB) framework for each exit layer. 4) We conduct experiments across five distinct datasets, and substantiate UBERT's proficiency in discerning the optimal threshold for early exit decisions, both in scenarios with a single exit layer and two exit layers. Our experimental validation demonstrates that, with appropriately chosen thresholds, there may be no necessity for more than two exits when exits are strategically positioned.

## 2    RELATED WORK

Previous works such as BranchyNet Teerapittayanon et al. (2016) use classification entropy at each attached exit after each layer to decide whether to infer the sample at the side branch. The decision is made to exit early at the side branch if the entropy is less than a given fixed threshold. Similar

architectures, SPINN Laskaridis et al. (2020) and SEE Wang et al. (2019b) make the decision of early exiting based on the estimated confidence measure provided by the side branch. They choose confidence as the probability of most likely class. Besides exiting early, works like FlexDNN Fang et al. (2020) and Edgent Li et al. (2019) focus mainly on the most appropriate NN depth. Other works such as Dynexit Wang et al. (2019a) focus on deploying the multi-exit NN in hardware. It trains and deploys the NN on Field Programmable Gate Array (FPGA) hardware while Paul *et al.* Kim & Park (2020) explains that implementing a multi-exit NN on FPGA board reduces inference time and energy consumption.

Pacheco *et al.* Pacheco et al. (2021) utilize both multi-exit NN and NN partitioning to offload mobile devices via multi-exit NNs. Similarly, EPNet Dai et al. (2020) learns when to exit considering the trade-off between overhead and accuracy in an offline fashion. Multi-exit NNs are being conformed in various other domains such as Image classification, ranking systems and NLP Bapna et al. (2020); Elbayad et al. (2020); Liu et al. (2021a); Xin et al. (2020a); Zhou et al. (2020). DeeBERT Xin et al. (2020a), ElasticBERT Liu et al. (2021a) and BERxiT Xin et al. (2021) are based on the transformer-based Vaswani et al. (2017) BERT model. BERxiT proposes an efficient fine-tuning strategy for the BERT model with attached exits. DeeBERT is obtained by training the exit points attached before the last module to the BERT backbone separately, whereas ElasticBERT is obtained by training all the exit points attached to the BERT backbone jointly. PABEE Zhou et al. (2020) is another multi-exit model that makes the exit decision based on the stability of the predictions after different exits. In a parallel vein, the Multiple Exiting (MuE) Tang et al. (2023) model employs a distinctive approach to determine early exits by assessing the similarity between consecutively learned hidden representations within the model's layers. Fei et al. (2022) on the other hand learns an imitation network to match the performance of deeper exits by learning a smaller network for image captioning. LEE Ju et al. (2021b), DEE Ju et al. (2021a) and UEE-UCB Hanawal et al. (2022) leverage the MAB framework to learn the optimal exit in EENNs. LEE and DEE mainly focus on learning optimal exits in image classification tasks, while UEE-UCB finds optimal exits for NLP tasks employing a pre-trained ElasticBERT Liu et al. (2021b) model. UEE-UCB does not need any label information but works under the assumption that the prediction of the intermediary layers follows strong dominance property Verma et al. (2019).

Our approach differs from past works as follows: 1) Unlike previous studies, our work primarily concerns determining the optimal threshold. 2) Our work is completely in an unsupervised online setup. 3) We do not make any structural assumptions like strong dominance property. 4) Using a single UCB configuration, we generalize adding multiple exits and discover the best thresholds for each exit simultaneously (in section 5) for two exits. The more general case of multiple exits is given in Appendix A.1. We compare against different early exiting models in table 2.

## 3 PROBLEM SETUP

### 3.1 EARLY-EXIT SETUP WITH ELASTICBERT

We consider classification tasks with a target class $\mathcal{C}$. We use a pre-trained ElasticBERT backbone with $l = 12$ transformer layers with classification heads attached to the output of specific layers that output scores for the target classes. We convert the scores into probability vectors by attaching a softmax layer. An input in ElasticBERT is processed sequentially through the intermediary layers outputting probability vectors at layers where classification heads are attached. Processing at an intermediary layer can stop and the sample can exit without passing through the following layers. We utilize the information from the side branches to decide if the sample exits at the intermediary level. More details on how we prepare a pre-trained ElasticBERT model could be found in the Appendix C.

Consider an intermediary layer $1 < p < l$. For an input $x$, let $\hat{P}_p(c)$ denote the estimated probability that $x$ belongs to class $c \in \mathcal{C}$ and $C_p$ denote the confidence in the estimate at the $p$th layer. We define $C_p$ as maximum of the estimated probability class, i.e., $C_p := \max_{c \in \mathcal{C}} \hat{P}_p(c)$. The decision to exit at a layer is made based on the value of confidence. For a given threshold, $\alpha$ if $C_p \geq \alpha$ the sample $x$ will be assigned a label $\hat{y} = \arg\max_{c \in \mathcal{C}}(\hat{P}_p(c))$. In this case, $x$ is not further processed, and *exits* the NN with a label $\hat{y}$. If $C_p < \alpha$, then the sample is processed to the next layer. We first consider the case where the exit is possible only at the $p$th intermediary layer, i.e., the sample can exit at layer $p$ with a label predicted by layer $p$, or it will get processed till the last layer $l$ and exits with label

predicted by layer $l$. We address the issue of exiting from more than one layer in Section 5. We denote the cost incurred from moving from the $p$th layer to the $l$th layer as $o$. It denotes the latency or computational cost of processing the sample between the layers $p$ and $l$.

At the $p$th layer, the confidence can be compared against one of the $k$ possible thresholds denoted by set $\mathcal{A}_p = \{\alpha_1, \alpha_2, \ldots, \alpha_k\}$. The goal is to identify the threshold which provides the best trade-off between loss in accuracy and latency cost. In the following, we define rewards obtained by each threshold and use the bandit framework to identify the optimal threshold.

### 3.2 MULTI-ARMED BANDIT SETUP

We treat the set of thresholds $\mathcal{A}_p$ as the set of actions. Following the terminology of MAB, we refer to them as arms. For any arm $\alpha \in \mathcal{A}_p$, we define the reward as follows

$$r(\alpha) = \begin{cases} 0 & \text{if } C_p \geq \alpha \\ C_l - C_p - o & \text{otherwise.} \end{cases} \tag{1}$$

For notational convenience, we write confidence gain in processing the sample from the $p$th layer to the $l$th layer as $\Delta C = C_l - C_p$, where $C_l$ is the confidence at the $l$th layer. Though confidence and latency are in different units, we add them after using a conversion factor. This factor is not explicitly shown as it can be absorbed into $o$. The reward could be interpreted as follows: if the learner decides to use the last layer, the reward is the gain in confidence minus the latency cost incurred, otherwise, the reward is zero. Then mean reward for arm $\alpha \in \mathcal{A}_p$ is

---

**Algorithm 1** UBERT

---
**Input:** $o, \gamma > 2$
**Initialize:** Play each threshold once. Observe $r(\alpha)$ and set $Q(\alpha) \leftarrow \mathbf{0}, N(\alpha) \leftarrow \mathbf{1}, \forall \alpha \in \mathcal{A}_p$.
**for** $t = |\mathcal{A}_p| + 1, |\mathcal{A}_p| + 2, \cdots$ **do**
    Observe an instance $x_t$

$$\beta_t \leftarrow \arg \max_{\alpha \in \mathcal{A}_p} \left( Q(\alpha) + \gamma \sqrt{\frac{\ln(t)}{N(\alpha)}} \right)$$

    Pass $x_t$ till layer $p$, apply threshold $\beta_t$ and observe $C_p$
    **if** $C_p \geq \beta_t$ **then**
        Infer at layer $p$ and exit
        **for** $\alpha \in \{\beta \in \mathcal{A} : \beta \leq \beta_t^p\}$ **do**
            $r_t(\alpha) \leftarrow 0, N_t(\alpha) \leftarrow N_{t-1}(\alpha) + 1$
            $Q_t(\alpha) \leftarrow \sum_{j=1}^t r_j(\alpha_j) \mathbb{1}_{\{\alpha_j = \alpha\}} / N_t(\alpha)$
        **end for**
    **else**
        Process and infer at the last layer. Observe $C_l$
        **for** $\alpha \in \{\beta \in \mathcal{A} : \beta \geq \beta_t^p\}$ **do**
            $r_t(\alpha) \leftarrow (C_l - C_p - o), N_t(\alpha) \leftarrow N_{t-1}(\alpha) + 1$
            $Q_t(\alpha) \leftarrow \sum_{j=1}^t r_j(\alpha_j) \mathbb{1}_{\{\alpha_j = \alpha\}} / N_t(\alpha)$
        **end for**
    **end if**
**end for**

---

$$\mathbb{E}[r(\alpha)] = \mathbb{E}[\Delta C - o \mid C_p < \alpha] \cdot P[C_p < \alpha] \tag{2}$$

Our goal is to find an arm with the highest mean reward. Since the setup is completely unsupervised, we depend on rewards to examine the progress of learning. Let $\alpha^* = \arg \max_{\alpha \in \mathcal{A}} \mathbb{E}[r(\alpha)]$ denote the optimal threshold. Consider a policy $\pi$ that selects threshold $\alpha_t \in \mathcal{A}_p$ in round $t$ based on past observations. We define cumulative regret of $\pi$ over $T$ rounds as

$$R(\pi, T) = \sum_{t=1}^T \mathbb{E}[r(\alpha^*) - r(\alpha_t)], \tag{3}$$

where the expectation is with respect to the randomness in the selection of thresholds induced by the past samples. A policy $\pi^*$ is said to be sub-linear if average cumulative regret vanishes, i.e., $R(\pi^*, T)/T \to 0$. Our objective is to develop a policy learning algorithm with a sub-linear regret guarantee.

## 4 ALGORITHM

We develop an algorithm named Unsupervised adaptive early exits in BERT named UBERT. Its pseudo-code is given in algorithm 1. The inputs to the algorithm are exploration constant $\gamma$ and latency factor $o$. For the first $|\mathcal{A}_p|$ samples, the algorithm plays each arm once. In the subsequent rounds, it plays the arm with the highest UCB index denoted as $\beta_t$. UCB indices are obtained by taking the weighted sum of the empirical average of the rewards $Q_t(\alpha)$ and the confidence bonuses

with $\gamma$ as the weight factor. If $C_p$ at the $p$th layer is larger than $\alpha_t$ then sample exits, otherwise, the sample is processed till the final layer incurring latency. Finally, the algorithm updates the number of pulls $(N(\beta_t))$ and empirical mean $(Q(\beta_t))$ of the played arm and the arms that belong to the set of side observations of the chosen exit as given in the algorithm. We obtain the set of side observations by analysing the behaviours of other arms for the given sample. For instance, if a sample exits at $p$th layer with confidence then it would have also exited for any arm smaller than the chosen arm.

Following the analysis of UCB1 Auer et al. (2002b), we show that the regret of UBERT is of $\mathcal{O}\left(\sum_{\alpha \in \mathcal{A}_p \setminus \alpha^*} \frac{log(n)}{\Delta_\alpha}\right)$ where $\Delta_\alpha = r(\alpha^*) - r(\alpha)$ denotes the optimality gap. For completeness, the proof outline is given in the Appendix B.1. Hence, UBERT acheives a sub-linear regret. The regret bound could be further improved by utilizing the side information. We explain it in the next section.

## 5    EXTENSION TO MULTIPLE EXITS

Samples not exiting from the first exit can be expected to have higher confidence in prediction as they are using more layers. Hence adding more exits is likely to show a gain in accuracy compared to the single-exit setup. However, now the thresholds need to be learned at all the exit points. We focus on the case with two exits to bring out the main ideas of the learning algorithm, and the generalization algorithm is given in the Appendix A.1. Also, in our experiments, we observed that all the samples exited before the second exit and only a small fraction passed to the next layers further justifying the restriction to the two-exits case (see figure 6).

Let $p$ and $q$ be the two intermediate exit layers satisfying $1 < p < q < l$. The confidence $C_p$ and $C_q$ at layers $p$ and $q$, respectively, are defined as in Sec. 3.1. At each exit layer, there are $k$ thresholds. We denote set of thresholds at exit layer $p$ and $q$ as $\mathcal{A}_p := \{\alpha_1^p, \alpha_2^p, \ldots, \alpha_k^p\}$ and $\mathcal{A}_q := \{\alpha_1^q, \alpha_2^q, \ldots, \alpha_k^q\}$, respectively. Note that $\mathcal{A}_p$ and $\mathcal{A}_q$ need not be the same.

Every sample is processed until the $p$th layer and the observed confidence $C_p$ is compared against a threshold $\alpha^p \in \mathcal{A}_p$. If $C_p \geq \alpha^p$ then it exits at layer $p$, otherwise the sample is processed till the $q$th layer and the confidence $C_q$ is observed. $C_q$ is then compared against a threshold $\alpha^q \in \mathcal{A}_q$. If $C_q \geq \alpha^q$, the sample exits at layer $q$, otherwise it is processed till the $l$th layer. If the sample is not inferred at the first exit layer, a cost denoted as $o_1$ is incurred. $o_1$ is defined as the latency/computational cost required to process the input from the $p$th layer till the $q$th layer. Similarly, if the sample is not inferred at the second exit layer, an additional cost denoted as $o_2$, is incurred. $o_2$ is the latency/computational cost in processing a sample from the $q$th layer to the $l$th layer. The total latency till the $l$th layer is $o_1 + o_2$ (see Eq. 4).

For any arm $(\alpha^p, \alpha^q) \in \mathcal{A}$, we define the reward as

$$r(\alpha^p, \alpha^q) = \begin{cases} 0 & \text{if } C_p \geq \alpha^p \\ C_q - C_p - o_1 & \text{if } C_p < \alpha^p, C_q \geq \alpha^q \\ C_l - C_p - o_1 - o_2 & \text{otherwise.} \end{cases} \tag{4}$$

For notational convenience we define $\Delta C_1 = C_q - C_p$ and $\Delta C_2 = C_l - C_p$. The mean reward for arm $(\alpha^p, \alpha^q) \in \mathcal{A}$ is then given as

$$\mathbb{E}[r(\alpha^p, \alpha^q)] = \mathbb{E}[\Delta C_1 - o_1 | C_p < \alpha^p, C_q \geq \alpha^q] \cdot P[C_p < \alpha^p, C_q \geq \alpha^q] + \\ \mathbb{E}[\Delta C_2 - o_1 - o_2 | C_p < \alpha^p, C_q < \alpha^q] \cdot P[C_p < \alpha^p, C_q < \alpha^q], \tag{5}$$

and the optimal arm is defined as the arm maximizing the mean reward and denoted as $(\alpha^{\star p}, \alpha^{\star q})$. For any policy $\pi$ that select the thresholds $(\alpha_t^p, \alpha_t^q)$ in round $t$ based on the past observations can be defined similar to 3 as $R(\pi, T) = \sum_{t=1}^{T} \mathbb{E}\left[r(\alpha^{\star p}, \alpha^{\star q}) - r(\alpha_t^p, \alpha_t^q)\right]$.

We extend UBERT for the case of two exits and refer to it as UBERT2. Its pseudocode is given in 2. In the two-exit case, the arm space is much larger compared to the case with a single exit, however, arms means are correlated which can be exploited to speed up the learning.

UBERT2 maintain the UCB index of each threshold pair in $\mathcal{A}$ and optimistically plays the one with the highest index in each round. When a pair $(\alpha^p, \alpha^q)$ is selected in a round and the sample exits at layer $p$, it is clear that the sample would have also exited at layer $p$ if any threshold smaller than

$\alpha^p$ is selected. Thus we get to observe not only the reward of the arm selected but also that of the other arms. This side observation is exploited by UBERT2 to improve learning performance. Though UBERT2 is similar to UCB-N Caron et al. (2012) and UCB-NE Hu et al. (2020) that exploit side-observations, their regret analysis does not apply to UBERT2 as a set of neighbours is not fixed–the neighbours of an arm depend on the confidence observed at the exit layers which can change from one sample to another. The following proposition helps to characterize the number of observations made for an arm.

---

**Algorithm 2** UBERT-2

---

1: **Input:** $o_1, o_2, \gamma > 2$
2: **Initialize:** Play each threshold once. Observe $r(\alpha^p, \alpha^q)$, $Q(\alpha^p, \alpha^q) \leftarrow \mathbf{0}$,
3: $N(\alpha^p, \alpha^q) \leftarrow \mathbf{1}, \forall (\alpha^p, \alpha^q) \in \mathcal{A}$.
4: **for** $t = |\mathcal{A}| + 1, |A| + 2, \cdots$ **do**
5:     Observe an instance $x_t$
6:     $(\beta_t^p, \beta_t^q) \leftarrow \arg \max\limits_{(\alpha^p, \alpha^q) \in \mathcal{A}} \left( Q(\alpha^p, \alpha^q) + \gamma \sqrt{\dfrac{\ln(t)}{N(\alpha^p, \alpha^q)}} \right)$
7:     Pass $x_t$ till layer $p$ and observe confidence $C_p$
8:     **if** $C_p \geq \beta_t^p$ **then**
9:       Infer at layer $p$ and exit
10:       **for** $(\alpha^p, \alpha^q) \in \{(\beta^p, \beta^q) \in \mathcal{A} : \beta^p \leq \beta_t^p \text{ and } \forall \beta^q \in \mathcal{A}_q\}$ **do**
11:         $r_t(\alpha^p, \alpha^q) \leftarrow 0, N_t(\alpha^p, \alpha^q) \leftarrow N_{t-1}(\alpha^p, \alpha^q) + 1$
12:         $Q_t(\alpha^p, \alpha^q) \leftarrow \sum_{j=1}^t r_j(\alpha_j^p, \alpha_j^q) \mathbb{1}_{\{(\alpha_j^p, \alpha_j^q)=(\alpha^p, \alpha^q)\}}/N_t(\alpha^p, \alpha^q)$
13:       **end for**
14:     **else if** Pass $x_t$ till layer $q$ and observe $C_q$ **then**
15:       **if** $C_q \geq \beta_t^q$ **then**
16:         Infer at layer $q$ and exit
17:         **for** $(\alpha^p, \alpha^q) \in \{(\beta^p, \beta^q) \in \mathcal{A} : \beta^p \geq \beta_t^p \text{ and } \beta^q \leq \beta_t^q\}$ **do**
18:           $r_t(\alpha^p, \alpha^q) \leftarrow C_q - C_p - o_1, N_t(\alpha^p, \alpha^q) \leftarrow N_{t-1}(\alpha^p, \alpha^q) + 1$
19:           $Q_t(\alpha^p, \alpha^q) \leftarrow \sum_{j=1}^t r_j(\alpha_j^p, \alpha_j^q) \mathbb{1}_{\{(\alpha_j^p, \alpha_j^q)=(\alpha^p, \alpha^q)\}}/N_t(\alpha^p, \alpha^q)$
20:         **end for**
21:       **end if**
22:     **else**
23:       Pass $x_t$ till the last year and infer. Observe $C_l$.
24:       **for** $(\alpha^p, \alpha^q) \in \{(\beta^p, \beta^q) \in \mathcal{A} : \beta^p \geq \beta_t^p \text{ and } \beta^q \geq \beta_t^q\}$ **do**
25:         $r_t(\alpha^p, \alpha^q) \leftarrow C_l - C_p - o_1 - o_2, N_t(\alpha^p, \alpha^q) \leftarrow N_{t-1}(\alpha^p, \alpha^q) + 1$
26:         $Q_t(\alpha^p, \alpha^q) \leftarrow \sum_{j=1}^t r_j(\alpha_j^p, \alpha_j^q) \mathbb{1}_{\{(\alpha_j^p, \alpha_j^q)=(\alpha^p, \alpha^q)\}}/N_t(\alpha^p, \alpha^q)$
27:       **end for**
28:     **end if**
29: **end for**

---

**Proposition 5.1** *Let $(i_t, j_t)$ denote the index of the threshold pair selected in round $t$ by UBERT2, and $C_p, C_q$ denote the associated confidence thresholds. Then probability that the reward of arm $(\alpha^p, \alpha^q) \in \mathcal{A}$ is observed in round $t$, denoted $P_t := P_t(\alpha^p, \alpha^q)$, is given by*

$$P_t = P(C_p \geq \alpha_{i_t}^p)\frac{i_t}{k} + P(C_p < \alpha_{i_t}^p, C_q \geq \alpha_{j_t}^q)\frac{(k-i_t)j_t}{k^2} + P(C_p < \alpha_{i_t}^p, C_q < \alpha_{j_t}^q)\frac{(k-i_t)(k-j_t)}{k^2}.$$

Note that the index $(i_t, j_t)$ selected in round $t$ is random as it depends on past observations. For any $\alpha := (\alpha^p, \alpha^q) \in \mathcal{A}$, let $\mathbb{E}[N_\alpha(T)]$ denote the expected number of times $\alpha$ is selected and $\mathbb{E}[O_\alpha(T)] = \mathbb{E}\left[\sum_{t=1}^T P_t\right]$ denote the expected number of times its reward is observed, where the expectation is over the distribution of all possible trajectories. It is clear that $\mathbb{E}[N_\alpha(T)] \leq \mathbb{E}[O_\alpha(T)]$.

Applying the standard UCB analysis that ignores the side-observation, the regret of UBERT2 over $T$ rounds is upper bounded by $R(\text{UBERT2}, T) \leq \mathcal{O}(k^2 \log T/\Delta^2)$ where $\Delta = r(\alpha^{\star p}, \alpha^{\star q}) - \min\limits_{(\alpha^p, \alpha^q) \neq (\alpha^{\star p}, \alpha^{\star q})} r(\alpha^p, \alpha^q)$ denotes the sub-optimality gap. However, as UBERT2 exploits the side observations, its regret upper bound is significantly smaller than this. Unfortunately, establishing the

exact bound is non-tractable as the set of neighbouring arms of each arm is not fixed. In the next section, we demonstrate that UBERT2 performs better compared to the UBERT that considers only the first exit and ignores the second exit but with added complexity.

## 6 EXPERIMENTS

In this section, we present the experimental setup, comprising three key phases:

**i) Pre-trained Backbone:** We utilize the ElasticBERT backbone, pre-trained using Masked Language Modeling (MLM) and Sentence Order Prediction (SOP) heads attached to each transformer layer of the BERT-base model. After training, we remove the heads, retaining the pre-trained backbone, as detailed in Liu et al. (2021b).

**ii) Fine-tuning (Pre-deployment):** Next, we strategically introduce task-specific exits into the pre-trained backbone. During this fine-tuning phase, we focus on supervised training to optimize exit weights. This prepares the model for deployment and testing on diverse datasets with distinct latent data distributions. Further details on the ElasticBERT training process can be found in Appendix C.

**iii) Unsupervised Online Threshold Learning (Post-deployment):** In the final stage, we employ the weights obtained in step (ii) to dynamically learn optimal thresholds in an unsupervised and online manner for evaluation tasks. This post-deployment step allows the model to autonomously adapt threshold values based on real-time data, enhancing adaptability in inference.

**Datasets:** We evaluated UBERT on five datasets covering four types of classification tasks. The datasets used for evaluation are: (1) **IMDb and Yelp:** IMDb is a movie review classification dataset and Yelp consists of reviews from various domains such as hotels, restaurants etc. For these two datasets, ElasticBERT was finetuned on **SST-2 (Stanford Sentiment classification)** dataset which is also a sentiment classification dataset. (2) **SciTail:** is an entailment classification dataset created from multiple questions from science and exams and web sentences. To evaluate UBERT on SciTail, ElasticBERT was finetuned on **RTE(Recognizing Textual Entailment)** dataset which is also an entailment classification dataset.

(3) **SNLI(Stanford Natural Language Inference:)** is a collection of human-written English sentence pairs manually labelled for balanced classification with labels *entailment*, *contradiction* and *neutral*. For evaluation of this dataset, ElasticBERT was finetuned on **MNLI(Multi-Genre Natural Language Inference)** which also contains sentence pairs as premise and hypothesis, the task is the same as for SNLI. (4) **QQP(Quora Question Pairs)** is a semantic equivalence classification dataset which contains question pairs from the community question-answering website Quora. We finetuned ElasticBERT on **MRPC(Microsoft Research Paraphrase Corpus)** dataset which also has a semantic equivalence task of a sentence pair extracted from online news sources. Details about the size of these datasets are in table 6. Observe from the table that the size of the dataset used for fine-tuning is much smaller as compared to the size of the corresponding evaluation dataset.

| Pos-Data | #Samples | Pre-Data | #Samples |
|----------|----------|----------|----------|
| IMDb | 25K | SST-2 | 68K |
| Yelp | 560K | SST-2 | 68K |
| SNLI | 550K | MNLI | 433K |
| QQP | 365K | MRPC | 4K |
| SciTail | 24K | RTE | 2.5K |

Table 1: This table provides some additional information for datasets. Pre-Data is the dataset used to fine-tune the ElasticBERT backbone before deployment for the corresponding Pos-Data (Dataset used after deployment) and #Samples is the number of samples in the dataset.

**Exit selection:** In our approach, we strategically position exit points at the 3rd and 6th layers of the ElasticBERT model, a decision informed by prior studies Scardapane et al. (2020); Bapna et al. (2020). Beyond the 6th layer, we refrain from incorporating exit points, primarily because confidence values tend to plateau, resulting in minimal gains (see Figure 2). In some instances, when samples traverse all the way to the final layer from the 6th layer, they experience a confidence loss (see Figure 5), due to overthinking similar to overfitting during training. Further insights into exit point selection are available in Appendix D.1.

**Choice of the action set:** The choice of the action set depends on the total number of output classes, denoted as $C$, within a given dataset. To ensure efficiency and avoid redundancy, we observe that any value in the action set below $1/C$ is extraneous. Consequently, we adopt a strategy of

choosing ten equidistant values ranging from $1/C$ to $1.0$ for the single exit case. For instance, in a binary classification scenario where the minimum confidence value is $0.5$, our action set becomes $\mathcal{A}_p = \{0.5, 0.55, 0.6, 0.65, \ldots, 0.95, 1.0\}$. Similarly, for the two-exit scenario, the action set choices are $\mathcal{A}_p, \mathcal{A}_q = \{0.5, 0.6, 0.7, 0.8, 0.9, 1.0\}$. We reduced the action set size for two-exit case to enhance computational efficiency.

Experiments were conducted on an NVIDIA RTX 2070 GPU, which was used for both ElasticBERT pre-training and obtaining evaluation dataset predictions. Pre-training typically took 6 to 8 hours, with a maximum of 24 hours for the MNLI dataset. Inference tasks averaged 1 to 2 hours, with the longest being approximately 6 hours for SNLI. After pre-training and obtaining confidence and predictions, evaluation tasks for both one and two exits were performed on an Intel Core i7 CPU with 16GB RAM, taking an average of 15 to 20 minutes to run the UBERT and UBERT2 algorithms

**Latency cost:** The user-defined latency cost, influenced by available computational resources, is a crucial factor in our approach. Leveraging the capabilities of the NVIDIA RTX 2070, with a peak FP32 performance of around 7.5 TFLOPS, we align the latency cost within the range of $0.1$/TFLOPS to $1$/TFLOPS, ensuring consistency with confidence gain units. Notably, we assume dedicated GPU usage for this purpose. The choice of latency can be tailored to specific tasks, striking a balance between accuracy and computational efficiency. Deeper layers incur progressively higher latencies, accommodating the additional time and resources required for samples to reach the second exit. For a single exit, we set the latency at $o = 0.1$, while in the two-exit scenario, we opt for $o_1 = 0.08$ and $o_2 = 0.04$, resulting in a cumulative latency of $0.12$ at the second exit. A detailed latency sensitivity analysis is available in Appendix D.4.

**Baselines:** We assess performance degradation by comparing against the baseline accuracy achieved by the BERT-base model's **Final Exit**. For reference, we also include the following models in our comparative analysis: **DeeBERT** and **ElasticBERT**, which utilize fixed confidence thresholds for early exiting decisions. **PABEE** employs prediction stability as a criterion for early exits, while **MuE** relies on hidden representation similarity for such decisions. To maintain consistency, we apply the MuE and PABEE approaches to the BERT-base model for comparison purposes. Hyperparameters for these baseline models remain consistent with their original implementations, and in the post-deployment phase, we adhere to fixed threshold values as learned by models during fine-tuning.

**Metrics:** As the measurement of runtime might not be stable even in the same environment, we utilize a new metric proposed in Tang et al. (2023) to evaluate the efficiency. It is called the expected time reduction rate which can be defined as:

$$1 - \frac{\sum_{i=1}^{l} x_i \times i}{\sum_{i=1}^{l} x_i \times l}$$

where $i$ is the $i$th layer in the network and $l$ is the total number of layers. $x_i$ is the number of samples exiting from layer $i$ (if an exit is attached). This metric observes the computation reduction ratio.

## 7 RESULTS

In Table 2, we provide the main results of this paper. We summarize the observations from each dataset in this section and explain the behaviour. We run each experiment 5- times. Each run includes an online feed of input samples to UBERT randomly rearranged. We provide average results over 5 runs. From our extensive evaluation in Table 2, it becomes evident that UBERT with two exits consistently outperforms all previous methods both in terms of accuracy and efficiency, thanks to its ability to adapt and select different thresholds at various layers of the model. Notably, we observe that UBERT2 opts for slightly lower thresholds for a deeper exit layer, a strategic choice driven by the observation that deeper layers exhibit a tendency to produce more accurate results and exit a higher proportion of samples.

This behaviour aligns with the fact that many samples, as illustrated in the confidence plots in Appendix (see figure 5), do not exhibit substantial confidence gains from 6th layer until the final layer (even most of the time confidence decreases). Failing to adjust or lower the threshold values for later exit layers would result in a majority of samples being processed until the final exit layer, thereby accumulating excessive inference time without considerable gain in accuracy. This saturation of confidence values in the deeper layers increases the likelihood of samples either exiting in the initial

| Model/Data | IMDb | | Yelp | | SciTail | | SNLI | | QQP | |
|---|---|---|---|---|---|---|---|---|---|---|
| | Acc | Time | Acc | Time | Acc | Time | Acc | Time | Acc | Time |
| **Final-exit** | 82.9 | 100 | 77.2 | 100 | 79.1 | 100 | 80.5 | 100 | 71.2 | 100 |
| **DeeBERT** | -6.1 | -43.3 | -3.9 | -58.9 | -0.5 | -8.3 | -2.5 | -28.3 | -6.7 | -50.1 |
| **ElasticBERT** | -4.5 | -45.9 | -3.4 | -62.9 | -0.2 | -14.7 | -2.2 | -30.1 | -4.7 | -52.5 |
| **PABEE** | -4.1 | -47.0 | -3.3 | -60.1 | -0.9 | -12.1 | -2.3 | -34.7 | -5.9 | -49.2 |
| **MuE** | -4.4 | -51.3 | -3.1 | -63.6 | -0.3 | -20.9 | -1.9 | -40.5 | -4.3 | -45.9 |
| **UBERT** | -5.7 | -52.5 | -4.5 | -61.3 | -0.2 | -9.2 | -1.6 | -46.0 | -1.8 | -17.9 |
| **UBERT-2** | **-3.9** | **-59.5** | **-3.1** | **-71.8** | **-0.1** | **-51.8** | **-1.1** | **-55.2** | **-0.3** | **-59.6** |

Table 2: Main results: The table provides the loss in accuracy (Acc) for different datasets and models as well as the decrease in inference time(Time). Observe that the reduction in time is in percentage reduction from the final exit case.

layers or exiting at the final layer. This justifies UBERT's choice of smaller thresholds for deeper layers. Previous methods like DeeBERT, ElasticBERT, PABEE, and MuE relied on fixed thresholds across layers and domains, resulting in either early exits from initial layers, leading to accuracy loss, or processing until the final layer, incurring higher inference costs. These scenarios were detrimental to model objectives. When evaluated on datasets with different latent distributions from the training set, these models often maintained a uniform threshold approach, potentially compromising accuracy and efficiency.

UBERT2 outperforms UBERT, which employs a single exit and a simple architecture. It excels on datasets resembling the training set distribution, where the 3rd exit gains higher confidence, leading to more early exits. This underscores the significance of exit placement and its impact on performance and inference time reduction. However, UBERT-2 comes with added complexity, hence not easy to analyse as comapred to UBERT. Further insights on the importance of UBERT with one exit are discussed in Appendix D.5

UBERT's dynamic threshold selection during inference, without requiring full retraining or fine-tuning, offers a substantial advantage. This adaptability significantly reduces inference time by 51% to 72%, while preserving accuracy. Notably, UBERT achieves this without relying on labeled data; instead, it makes real-time threshold decisions based on evolving data distributions. By utilizing available side information, UBERT and UBERT2 converge to optimal thresholds after just a few thousand samples, as discussed in Appendix E. Additionally, in Appendix D.3, we empirically demonstrate that UBERT2 consistently outperforms fixed thresholds, including those chosen by previous methods.

# 8 CONCLUSION

In this work, we have introduced an innovative online algorithm designed to determine whether to terminate the processing of a sample at the intermediate layers of a neural network or at the final layer. Our algorithm effectively minimizes the inference time of pre-trained ElasticBERT model while making only minor compromises in terms of accuracy. Notably, our approach operates in an unsupervised manner, eliminating the need for labeled data to optimize the delicate balance between accuracy loss and reduced latency.

In this work, we could only argue that UBERT2 exploits side information to give better performance even though it has to search over a large number of arms to find the optimal arm. It is interesting to quantify the regret of UBERT2 and how to further exploit the side information to improve its regret performance. To circumvent the issue of complexity in a dding multiple exits, a strategic placement of exits is required.

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

# A APPENDIX

## A.1 EXTENSION TO MULTIPLE EXITS

In this section, we consider multiple exits and simultaneously learning the optimal thresholds for $E$ exits for $1 \leq E < L$. Adding multiple exits will surely show a gain in accuracy as compared to one or two-exit setup. We denote the set $\{1, 2, \ldots E\}$ as $[E]$.

For the intermediate exit $l \in [E]$, let $C_l$ be the confidence at exit $l$ and are defined as in Sec. 3.1. We denote the action set at exit $l$ as $\mathcal{A}_l = \{\alpha_1^l, \alpha_2^l, \ldots, \alpha_k^l\}$. Since now we are simultaneously choosing the thresholds for $E$ exits, the action set changes to $\mathcal{A} = \mathcal{A}_1 \times \mathcal{A}_2 \times \ldots \times \mathcal{A}_E$.

The sample is processed till first exit layer and $C_1$ is observed then if $C_1 \geq \alpha^1$ the sample exits the backbone else it is processed to next exit layer. The sample is processed until $C_l \geq \alpha^l$ for $l \in [E]$ and the sample exits the backbone. Else, the sample is inferred at the final layer with confidence $C_L$. If the sample is processed till exit $l$ then the total latency occurred is $o_1 + o_2 + \ldots + o_l$ In the multiple exits case, for any arm $\alpha = (\alpha^1, \alpha^2, \ldots, \alpha^E)$ the reward function will be defined as:

$$r(\alpha) = \begin{cases} 0 & \text{if } C_1 \geq \alpha^1 \\ \vdots & \\ C_l - C_1 - o_1 - o_2 - o_3 \ldots o_l & \text{if } C_1 < \alpha^1, C_2 < \alpha^2 \ldots C_l \geq \alpha^l \\ \vdots & \\ C_L - C_1 - o_1 - o_2 - \ldots o_E & \text{if } C_1 < \alpha^1, C_2 < \alpha^2 \ldots C_E < \alpha^E \end{cases} \quad (6)$$

The mean reward for arm $\alpha = \{\alpha^1, \alpha^2, \ldots, \alpha^E\}$ is then given as:

$$\mathbb{E}[r(\alpha)] = \sum_{l=1}^{E} \mathbb{E}[C_l - C_1 - o_1 - \ldots - o_l | C_1 < \alpha^1, \ldots, C_{l-1} < \alpha^{l-1}, C_l \geq \alpha^l]$$
$$. P[C_1 < \alpha^1, \ldots, C_{l-1} < \alpha^{l-1}, C_l \geq \alpha^l] + \mathbb{E}[C_L - C_1 - o_1 - \ldots - o_E | C_1 < \alpha^1, \ldots, C_E < \alpha^E]$$
$$. P[C_1 < \alpha^1, \ldots C_E < \alpha^E] \quad (7)$$

## A.2 ALGORITHM

In this section, we provide the general algorithm for any number of exits. The algorithms UBERT and UBERT-2 in the main paper are just the cases when we have one and two exits. Also here $\alpha = (\alpha^1, \alpha^2, \ldots, \alpha^E)$ where $l$th component of the vector represents the threshold for the $l$th exit. The input to the algorithm is the overhead cost $o_l$ for processing the input from exit layer $l - 1$ to exit layer $l$ and exploration parameter $\gamma$. In every iteration, a vector of size $E$ is chosen (line 5) based on the UCB index and the $l$th component is compared against the confidence at $l$th exit point until the confidence is above the chosen threshold. The reward is then observed based on the layer from which the sample exited. If the sample's confidence does not meet the threshold for the intermediate exits, it is then processed till the final layer which infers the sample with a reward. Suppose there are $K$ thresholds for every exit layer and there are $E$ exits attached, then the size of the action set will be $|\mathcal{A}| = K^E$. There is an exponential increase in the size of the action set with an increasing number of exits which in turn requires more samples to saturate to optimal exit. To circumvent this issue, we utilize the available side information. The algorithm performs multiple updates in a single iteration by analysing the structure of the reward function as well as the action set. The arms that will get updated (line 9) in every iteration depend on the exit at which the sample is inferred as well as the chosen arm. By using side observations, the algorithm learns faster without accumulating significant regret. The algorithm in the main paper for two exits looks more complex as the while loop in line 7 of UBERT-E algorithm is unrolled in UBERT-2 for better understanding.

# B PROOF FOR PROPOSITION 1

We will complete the proof according to the confidence observed at different exit layers.
If the sample exits at the first exit layer then the set $S_1 = \{(\gamma^p, \gamma^q) : \gamma^p \leq \alpha_{i_t}^p \text{ and } \forall \gamma^q \in \mathcal{A}_q\}$ will

---

**Algorithm 3** UBERT-E

---

1: **Input:** $o_i \forall i \in [E], \gamma > 2$
2: **Initialize:** Play each threshold once. Observe $r(\alpha), Q(\alpha) \leftarrow \mathbf{0}, N(\alpha) \leftarrow \mathbf{1}, \forall \alpha \in \mathcal{A}$.
3: **for** $t = |\mathcal{A}| + 1, |A| + 2, \cdots$ **do**
4:     Observe an instance $x_t$
5:     $\beta_t \leftarrow \arg\max\limits_{\alpha \in \mathcal{A}} \left( Q(\alpha) + \gamma\sqrt{\dfrac{\ln(t)}{N(\alpha)}} \right)$
6:     $l = 1$ and $o_1 = 0$
7:     **while** $C_l \geq \beta_t^l$, and $l \in [E]$ **do**
8:         Infer at layer $l$ and exit
9:         **for** $\alpha \in \{\beta \in \mathcal{A} : \beta^p > \beta_t^p \quad \forall p \in [l-1], \ \beta^l \leq \beta_t^l \text{ and } \forall \beta^q \text{ where } q > l\}$ **do**
10:             $r_t(\alpha) \leftarrow C_l - C_1 - o_1 - o_2 - \ldots - o_l, \ N_t(\alpha) \leftarrow N_{t-1}(\alpha) + 1$
11:             $Q_t(\alpha) \leftarrow \sum_{j=1}^{t} r_j(\alpha_j)\mathbb{1}_{\{\alpha_j = \alpha\}}/N_t(\alpha)$
12:         **end for**
13:         $l = l + 1$
14:     **end while**
15:     **if** $l = L$ **then**
16:         Pass $x_t$ till the last year and infer. Observe $C_L$.
17:         **for** $\alpha \in \{\beta \in \mathcal{A} : \beta^p > \beta_t^p \ \forall p \in [L-1]\}$ **do**
18:             $r_t(\alpha) \leftarrow C_L - C_1 - o_1 - o_2 - \ldots - o_E, \ N_t(\alpha) \leftarrow N_{t-1}(\alpha) + 1$
19:             $Q_t(\alpha) \leftarrow \sum_{j=1}^{t} r_j(\alpha_j)\mathbb{1}_{\{\alpha_j = \alpha\}}/N_t(\alpha^p, \alpha^q)$
20:         **end for**
21:     **end if**
22: **end for**

---

be updated. Now

$$P((\alpha^p, \alpha^q) \in S_1) = |S_1|/k^2 = i_t/k$$

Hence probability that $(\alpha^p, \alpha^q)$ will get updated at first intermediate layer is $P(C_p \geq \alpha_{i_t}^p) \cdot i_t/k$.
If the sample exits at the second exit layer then the set $S_2 = \{(\gamma^p, \gamma^q) : \gamma^p > \alpha_{i_t}^p \text{ and } \gamma^q \leq \alpha_{j_t}^q\}$
will get updated. Similarly,

$$P((\alpha^p, \alpha^q) \in S_2) = |S_2|/k^2 = (k - i_t)j_t/k^2$$

Hence probability that $(\alpha^p, \alpha^q)$ will get updated at second intermediate layer is $P(C_p < \alpha_{i_t}^p, C_q \geq \alpha_{j_t}^q) \cdot (k - i_t)j_t/k^2$.
Finally, if the sample exits at the final layer, then the set $S_3 = \{(\gamma^p, \gamma^q) : \gamma^p > \alpha_{i_t}^p \text{ and } \gamma^q > \alpha_{j_t}^q\}$
will get updated. So,

$$P((\alpha^p, \alpha^q) \in S_3) = |S_3|/k^2 = (k - i_t)(k - j_t)/k^2$$

Hence the probability that $(\alpha^p, \alpha^q)$ will be updated at final layer is $P(C_p < \alpha_{i_t}^p, C_q < \alpha_{j_t}^q) \cdot (k - i_t)(k - j_t)/k^2$. Hence probability that the reward of arm $(\alpha^p, \alpha^q) \in \mathcal{A}$ is observed in round $t$, denoted $P_t := P_t(\alpha^p, \alpha^q)$, is given by $P_t = P(C_p \geq \alpha_{j_t}^p)\frac{i_t}{k} + P(C_p < \alpha_{j_t}^p, C_q \geq \alpha_{j_t}^q)\frac{(k - i_t)j_t}{k^2} + P(C_p < \alpha_{i_t}^p, C_q < \alpha_{j_t}^q)\frac{(k - i_t)(k - j_t)}{k^2}$

### B.1 Upper bound on Regret of UBERT

**Theorem B.1** *For any $\gamma > 1$, the regret of UBERT with $K$ arms in the action set after $n$ rounds is given as:*

$$R(UBERT, n) \leq 4\gamma \sum_{\alpha \neq \alpha^*} \frac{\log(n)}{\Delta_\alpha} + (\pi^2/3 + 1)\sum_{\alpha \neq \alpha^*} \Delta_\alpha \tag{8}$$

*where $\Delta_\alpha = r(\alpha^*) - r(\alpha)$*

The proof is very similar to the classical UCB1 Auer et al. (2002a) and follows the same lines with noting the regret in round $t$ as

$$R_t = r(\alpha_t) - r(\alpha^*)$$

$r(\alpha)$ is a bounded quantity by definition and more specifically $r(\alpha) \in [-1 - o, 1]$, where $o$ is the latency cost accumulated over all the exits.

One can provide even better bounds than given in eq. 8, after taking the side observations into account that are available in each round and using proposition 5.1.

## C  TRAINING AN ELASTICBERT MODEL

In order to evaluate the performance of UBERT, it is essential to have a pre-trained multi-exit neural network that aligns with a similar task. This section outlines the procedure for creating a specific multi-exit deep neural network (DNN) model, which will serve as the benchmark for assessing UBERT's effectiveness. To construct this multi-exit DNN, we start with the ElasticBERT-base model, which is built upon the BERT-base architecture. We initiate the training process by attaching an exit after each layer of the BERT-base model and employing a joint loss function that combines masked language modelling and sentence order prediction across all exits. This training occurs on a substantial text corpus. Following this training phase, we remove the heads of the exits, specifically the Masked Language Modeling and Sentence Order Prediction components, leaving behind the ElasticBERT-base model's backbone and the associated learned weights. This resulting model possesses the capability to generate language representations that are particularly well-suited for early-exit scenarios. It forms the foundation for our evaluation and comparison with UBERT across a range of applications, including sentiment classification, natural language inference, paraphrasing tasks, and textual entailment classification.

Following the preparation of the ElasticBERT backbone, we proceed to attach task-specific exits, including classification heads, after each layer of the backbone. To provide an overview, let's consider a sentence-level task like sentiment classification, where we utilize a special token denoted as *[CLS]* to facilitate the learning of sentence-level representations. Each token's output representation is then connected to a classification head positioned after each attention layer. The primary aim of these classification heads is to generate representations that can be effectively compared to the task-specific labels. For example, in the case of binary textual entailment classification tasks, where the labels are binary (positive/negative), we typically employ the binary cross-entropy loss as our preferred loss function. This loss function allows the classification head to generate

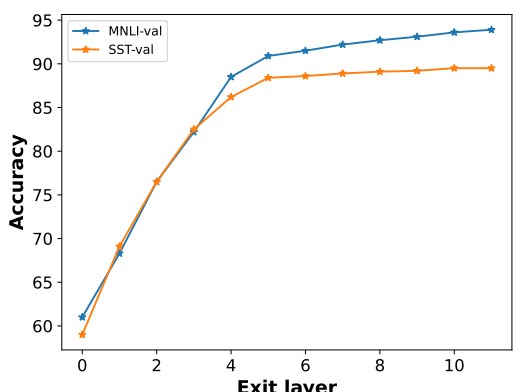

Figure 2: The validation set accuracy of the datasets MNLI and SST-2 that were used to place exits at different points in the backbone.

scores by transforming the d-dimensional vector representation of the *[CLS]* token using learnable weights. Our fine-tuning process involves training the ElasticBERT backbone on datasets that align with similar task types. As an illustration, we leverage the ElasticBERT model, which was initially pre-trained on the SST-2 dataset for sentiment classification, to evaluate datasets like IMDb and Yelp, both of which involve positive/negative review classification tasks.

According to the hyperparameter selections made in ElasticBERT, the ElasticBERT model is fine-tuned on each dataset for 5 iterations. Every 50 steps, the model is checkpointed. The model with the highest average accuracy across all exits is the final model.

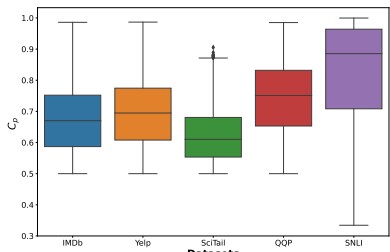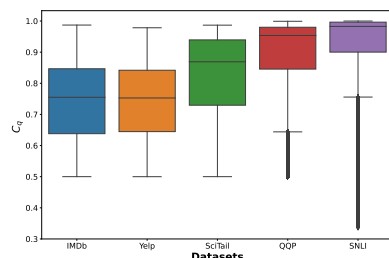

Figure 3: $C_p$ (confidence values at $p$th exit) and $C_q$ (confidence values at $q$th exit) for different datasets

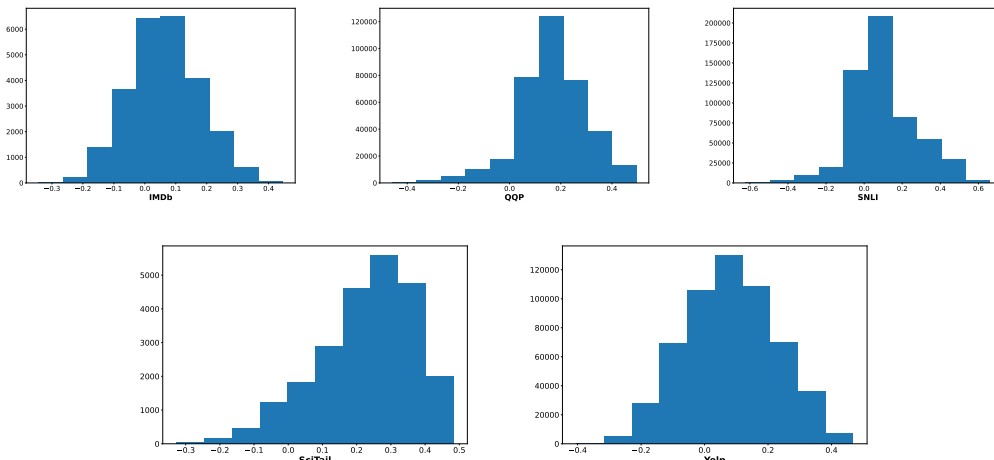

Figure 4: **(y-axis represents number of samples and x-axis is the gain in confidence)** Confidence gain ($\Delta C$) from first exit to final layer for UBERT. There is a gain in confidence for most of the samples across all the datasets (except IMDb which suggests exiting many samples from the first exit)

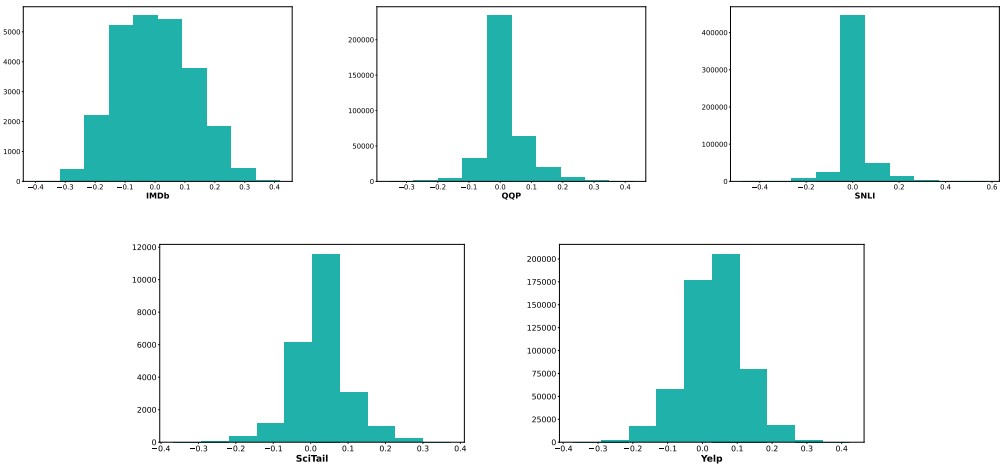

Figure 5: **(y-axis represents number of samples and x-axis is the gain in confidence)** Confidence gain ($\Delta C_2$) from second exit to final layer. Observe that from the second exit to the final exit, many samples even lose confidence and only a few samples observe a gain confidence (that also a minimal gain)

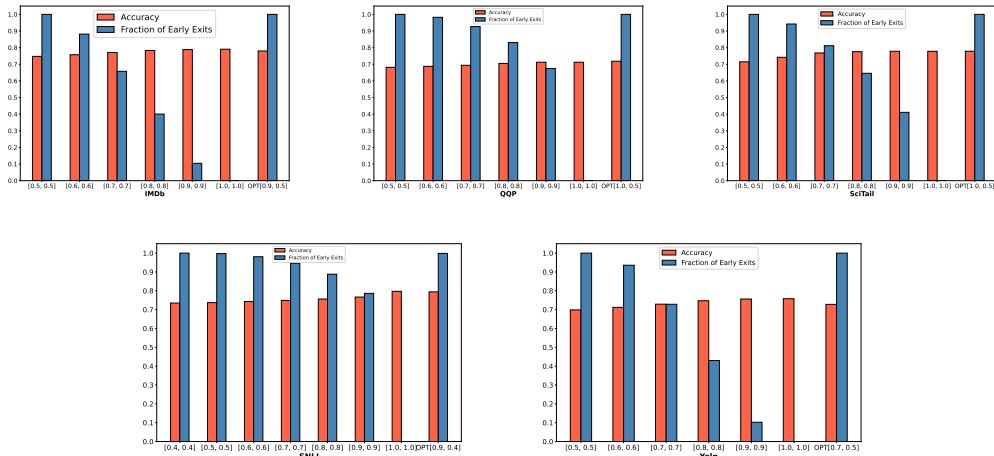

Figure 6: Comparison of UBERT-2 with fixed threshold values where **OPT** means the optimal thresholds chosen for both exits

# D EXPERIMENTS (CONTD..)

## D.1 CHOICE OF EXIT LAYERS

In our experimentation, we strategically positioned exit layers at the 3rd and 6th layers of the model architecture. This decision was informed by an observation made during the fine-tuning process, specifically when assessing the behavior of accuracy on the validation subset of the dataset used for fine-tuning. To ensure a robust evaluation, we reserved 20% of the data for validation purposes. Upon fine-tuning ElasticBERT on the training split of the dataset, we noticed a flattening of the accuracy curve after the 6th layer, a trend visually represented in Figure 2. For most of the datasets, we conducted this validation, accuracy tended to stabilize after 6th layer. Also, the 3rd layer also has a sharp increase in accuracy making it a suitable choice to attach exit. Subsequently, we decided to place the exit layers at the 3rd and 6th layers, anticipating that this pattern of accuracy improvement would carry over to evaluation datasets as well.

Also, from an application point of view in an edge-cloud co-inference setup, attaching exits to deeper layers requires us to incorporate more layers on the edge device making the whole model less efficient. More details on how to attach exits can be found in Scardapane et al. (2020); Bapna et al. (2020).

## D.2 CONFIDENCE ANALYSIS

In Figure 3, we provide an analysis of the confidence levels observed at the first exit layer. Notably, we find that the optimal threshold for the SciTail dataset is relatively high, at 0.85. This higher threshold effectively compels most samples to continue to the final layer, primarily because the confidence levels at the first exit layer tend to be modest, typically falling within the range of 0.55 to 0.65, as depicted in Figure 3. Furthermore, Figure 4 reveals that the gain in confidence from the first exit layer to the final layer is substantial, providing a compelling rationale for UBERT to exit fewer samples at the first layer. For other datasets and the one-exit scenario, the threshold is contingent on the confidence levels observed at the first exit layer. As an illustration, consider the IMDb and Yelp datasets, where we observe slightly lower optimal thresholds. This observation aligns with the fact that the gain in confidence is comparatively smaller for these datasets, encouraging UBERT to exit a larger proportion of samples at the first exit.

Moreover, we notice that for most datasets, all samples exit at the second layer, a phenomenon that can be elucidated by referring to Figure 3 and Figure 5. These figures clearly demonstrate that, after progressing from the second exit to the final layer, samples do not exhibit significant gains in confidence. In fact, for many datasets, confidence levels either remain stagnant or even decrease when processed until the final layer. This behavior is attributed to overfitting tendencies. Given the high

|  | **IMDb** | | | **Yelp** | | | **SciTail** | | |
| **LatF** | **Acc** | **ExitP** | **Opt Thr** | **Acc** | **ExitP** | **Opt Thr** | **Acc** | **ExitP** | **Opt Thr** |
| **0.00** | 0.821 | 0.152 | 0.75 | 0.742 | 0.371 | 0.75 | 0.778 | 0.001 | 0.90 |
| **0.05** | 0.787 | 0.541 | 0.65 | 0.733 | 0.531 | 0.70 | 0.777 | 0.014 | 0.95 |
| **0.10** | 0.773 | 0.705 | 0.60 | 0.724 | 0.668 | 0.65 | 0.775 | 0.041 | 0.80 |
| **0.15** | 0.758 | 0.869 | 0.55 | 0.716 | 0.787 | 0.60 | 0.766 | 0.189 | 0.70 |
| **0.20** | 0.747 | 1.000 | 0.50 | 0.709 | 0.896 | 0.55 | 0.75 | 0.426 | 0.65 |

Table 3: This table compares the values of various latency costs and details how they affect accuracy, the fraction of early exiting samples, and the related optimal threshold (Opt Thr).

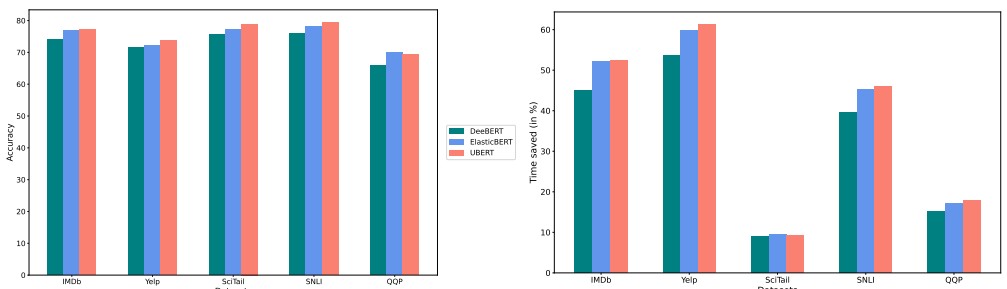

Figure 7: The left figure shows the accuracy comparison and the right shows the time saved when DeeBERT and ElasticBERT are applied in the same setup as UBERT.

confidence and limited confidence gain at the second exit, UBERT-2 efficiently guides all samples to exit at the second layer.

These observations provide valuable insights into the dynamic behavior of UBERT and UBERT-2 across various datasets and exit layer configurations.

### D.3 COMPARISON WITH FIXED THRESHOLDS (UBERT-2)

In Figure 6, we present a comprehensive comparison between the two-exit scenario and various baseline models, specifically DeeBERT and ElasticBERT. Notably, both of these baseline models employ fixed thresholds consistently across all exit layers. In this section, we experimentally validate that for any value of thresholds, if they are fixed across all exit points, UBERT-2 will always outperform them.

A striking observation is that, across most datasets, our UBERT-2 model outperforms these baselines in terms of the proportion of samples exiting early, all while maintaining competitive accuracy levels. Moreover, for certain datasets, UBERT-2 even achieves higher accuracy rates while ensuring a greater fraction of samples exit early compared to the fixed threshold counter parts. These findings strongly suggest that the adoption of different threshold values for distinct layers is a crucial strategy. Deeper layers, characterized by higher confidence and more accurate predictions, benefit from lower thresholds, enabling them to exit samples early. In contrast, initial exits are better suited with higher thresholds, ensuring that only highly confident samples exit early, given their lower accuracy levels.

It's worth noting that this nuanced threshold adaptation is a feature unique to UBERT-2, setting it apart from previous works, both of which rely on fixed thresholds across all exits. This adaptive approach contributes to UBERT-2's superior performance in terms of balancing accuracy and the fraction of early exits, highlighting its potential as an efficient and accurate early-exit model.

### D.4 SENSITIVITY ANALYSIS ON DIFFERENT LATENCY COSTS

In Table 3, we provide a comprehensive overview of our findings regarding the impact of various latency factors (LatnF) on the optimal thresholds (Opt Thr) for the one-exit case, their corresponding accuracy (Acc), and the fraction of samples exiting early (ExitP). Our observations consistently reveal a clear trend across all datasets: as the latency factor increases, there is a simultaneous decrease in

accuracy coupled with a rise in the proportion of samples exiting early. This phenomenon can be attributed to the fact that higher latency costs lead to the derivation of lower optimal thresholds, subsequently compelling more samples to exit early due to diminished confidence, thereby contributing to lower accuracy.

It is worth highlighting a notable observation: even when the latency factor is set to zero, some samples still exit before reaching the final layer. This occurrence suggests that certain samples do not experience a substantial boost in confidence as they progress through the network. Notably, in the case of the SNLI and Yelp datasets, an intriguing finding emerges, indicating that a substantial portion (37%) of samples exit early when there is no latency cost. This observation implies that certain samples may experience a decrease in confidence during processing until the final layer, a phenomenon that may be indicative of overthinking (analogous to overfitting in training).

### D.5 UBERT'S SIGNIFICANCE

As illustrated in Table 2, UBERT exhibits superior performance in a specific dimension, either achieving a marginal decrease in accuracy or significantly reducing inference time. This outcome aligns with the inherent design of UBERT, where an exit is strategically placed at the third layer. Consequently, each data sample undergoes inference either at this intermediate layer or at the final layer. The distribution of samples between these exits plays a crucial role: if a substantial proportion exits at the third layer, UBERT's accuracy is impacted; conversely, if most exit at the final layer, inference time is affected. Nevertheless, UBERT distinguishes itself by not conducting inference after every layer, thereby mitigating the inference cost compared to earlier methodologies.

In Table 2, we refrain from directly reducing the inference cost. Instead, we employ a comparative evaluation of UBERT against ElasticBERT and DeeBERT under similar conditions, where we equip DeeBERT and ElasticBERT with a single exit and a fixed threshold determined through validation data during fine-tuning in Figure 7.

This necessitated an additional exit point at the sixth layer, resulting in the UBERT-2 configuration. This augmentation has proven to be highly beneficial, leading to improvements in both accuracy and inference speed. The placement of this deeper exit allows a substantial number of samples to reach a confidence threshold and undergo inference at an earlier stage in the network. Consequently, this modification has a positive impact on both accuracy and inference time. Notably, our empirical observations indicate that, with the exception of the SNLI dataset, all the samples exit the deep neural network before reaching the final layer when utilizing the UBERT-2 configuration. In the case of SNLI, only a negligible 0.2% of samples continue to the final layer during the inference process. Given that all samples exit from either the third or sixth layer, there is no compelling need for the addition of a third exit to the network backbone.

## E REGRET PERFORMANCE

### E.1 ONE-EXIT

In figure 8, we compare the cumulative regret of UBERT with different baselines that are DeeBERT and ElasticBERT as fixed thresholds. Each experiment is run 5 times, and the estimated cumulative regret is plotted with 95% confidence ranges. Each run includes an online feed of input examples that are randomly rearranged and fed to the algorithm. We take into account the following threshold while benchmarking the binary classification task: $\alpha = 0.5, 0.8, 0.9$ and $1.0$. $\alpha = 0.5$ corresponds to the case when all instances exit at the intermediate layer. $\alpha = 1.0$ corresponds to the case when all instances are processed till the last exit. In the SNLI dataset case, we have also used $\alpha = 0.3$ for benchmarking instead of $\alpha = 0.5$, as now all samples will exit early at $\alpha = 0.3$. In plots, the $x$-axis represents the number of input samples fed to the algorithm sequentially, for large datasets such as IMDb and Yelp we have every 50th sample respectively on $x$-axis. We observe that the cumulative regret saturates with the increasing number of input samples. For the IMDb plot, the cumulative regret of UBERT is better compared to other baselines except for the case of $\alpha = 0.5$ on the IMDb dataset. This could be understood as the $\alpha = 0.5$ is close to the optimal threshold. Similar to this, for the datasets QQP and SciTail, UBERT is near one of the many fixed thresholds that are close to the optimal threshold, so their regret for every sample is very small. UBERT initially explores

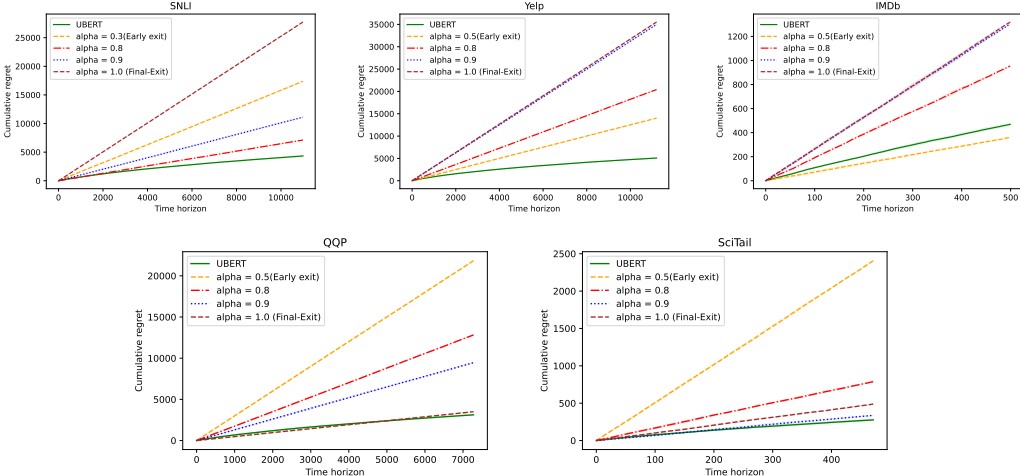

Figure 8: Cumulative regret curves for one exit case

non-optimal thresholds before gradually converging towards the optimal threshold and exploits it after a few thousand iterations.

Our results indicate that UBERT effectively identifies the optimal threshold for a significant portion of samples. It's important to note that UBERT does not rely on dataset ground truth labels for its threshold decisions; instead, these ground truths are solely employed to assess model performance and accuracy.

Another noteworthy observation is the substantial difference in dataset sizes between the finetuning and evaluation phases. Even when the ElasticBERT backbone is fine-tuned on a considerably smaller dataset with a similar task, UBERT consistently manages to determine an appropriate threshold. This resilience to dataset size variations and differences between finetuning and evaluation data stems from UBERT's ability to find the threshold while solely utilizing confidence values at the intermediate and final layers.

### E.2 TWO-EXIT

In figure 9, we compare the cumulative regret of UBERT-2 with different baselines which are fixed thresholds.

The experimental setup is the same as the one exit case. Notice that $(\alpha^p, \alpha^q) = (0.5, 0.5)$ will exit all the samples at the first exit layer. Also $(\alpha^p, \alpha^q) = (1.0, 1.0)$ will exit all the samples from the final layer. Observe that for SNLI and IMDb datasets, the regret for one pair of fixed thresholds is lower than UBERT-2. This effect appears as the reward for these actions might be very close to the optimal action. This effect can be removed by increasing the dataset size. Except for these datasets, the regret for UBERT-2 was lower than the fixed thresholds. The cumulative regret in the two-exit case takes more rounds to saturate as it has a large action set to choose the optimal action.

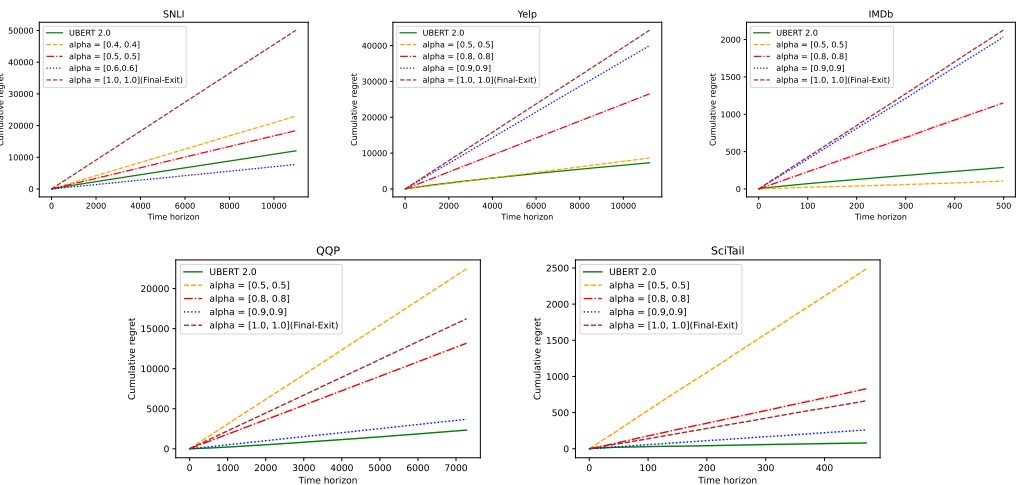

Figure 9: Cumulative regret curves for two-exit case

