# OpenReview forum: "UBERT: Unsupervised adaptive early exits in BERT"
_ICLR.cc/2024/Conference — ICLR 2024 Conference Withdrawn Submission_

### Official Review · Reviewer_isHH · 2023-10-26

**Soundness:** 2 fair
**Presentation:** 2 fair
**Contribution:** 2 fair
**Rating:** 3
**Confidence:** 4

**Summary:**

This paper studies the settings for early-exiting thresholds, which determines which samples will be output at each early exit during inference. This is an interesting topic in both NLP and CV. An algorithm is proposed to decide the confidence thresholds without groud-truth labels. Experiments on NLP tasks show that the method outperforms existing early-exiting approaches. However, there are some overclaims, and important references are missing. Moreover, the experiment results are not convincing enough (see weakness).

**Strengths:**

1. The studied topic is interesting, and the motivation is clearly explained;
2. The method is technically sound.

**Weaknesses:**

1. **Overclaim**. The authors claimed that existing methods usually use labeled datasets to determine the early-exiting thresholds, and the proposed method removes the need for ground-truth labels. This is not correct. Maybe the cited baselines in NLP do need ground-truth labels. But if the authors pay attention to the dynamic models in the CV field (see the second part, **Missing references**), they can find that the decision of confidence thresholds can purely rely on the confidence distribution on the training/validation set. Specifically, one can decide the ratio of samples exiting at different exits, and solve the threshold based on the confidence scores of each exit without touching the ground-truth labels. In summary, the main contribution claimed by the authors, may not hold.

2. **Missing references**. It is recommended that the authors compare their method with the aforementioned strategies in the CV field [1,2,3].

3. **Inconvincing experiments**. In Tab. 2, the proposed method is compared with other baselines at a **fixed** computational cost. However, the main advantage of dynamic early exiting is one can adjust the thresholds for different computational budgets (see the smooth curves in [1,2,3]. It is kindly suggested that the proposed method is compared with the "ratio -> threshold" pipeline in the CV field.

[1] Huang et al, Multi-Scale Dense Networks for Resource Efficient Image Classification.

[2] Yang et al, Resolution Adaptive Networks for Efficient Inference.

[3] Han et al, Dynamic Perceiver for Efficient Visual Recognition.

**Questions:**

See weaknesses.

---

> ### Author Response · Authors · 2023-11-15
> **Rebuttal**
>
> We thank the reviewer for the feedback!
>
> Q1: The authors claimed that existing methods usually use labeled datasets to determine the early-exiting thresholds, and the proposed method removes the need for ground-truth labels. This is not correct............
>
> A1: To be clear, we use both the terms online and unsupervised, hence both the combinations have not been tried early. The methods given by the reviewer work in an offline manner.
>
> As per our understanding, recent works such as 'DeeBERT,' 'ElasticBERT,' 'FastBERT' in NLP models, alongside image captioning models like 'DeeCap,' 'Multiple exiting,' and image classification models such as 'BranchyNet,' 'AdaEE,' among others, predominantly involve tuning the threshold for early exiting decisions based on the validation dataset, hence being commonly referred to as using this approach.
>
> While previous image-related works have indeed explored learning optimal threshold values based on confidence measures, these values are typically adjusted concerning the confidence distribution within the validation data, which mirrors the distribution of the training dataset in all the referenced studies. The testing and validation datasets share a similar distribution to the training dataset, ensuring the model operates within analogous environments. Also, there is a strong assumption of constant exit probability distribution across all the exits, which is not true in most cases as deeper exit will have higher exiting probability than initial exits as explained in our paper in Appendix D.1.
> In contrast, our work specifically addresses post-deployment scenarios where a pre-trained DNN encounters testing with datasets possessing differing distributions from those used during training. This variance leads to a distinct confidence distribution during testing. For instance, a model trained for sentiment classification (SST-2) may face testing in movie review classification (IMDb) (we do not use any ground-truth values from IMDb dataset hence calling it unsupervised), signifying a shift in the underlying dataset distribution. Similarly, in image analysis, post-deployment testing encounters distribution shifts due to image distortions caused by environmental factors like time of day and weather, as discussed in AdaEE. This necessitates adaptive threshold selection. Also, budget setups can force some samples to exit early due to finite budget settings and make the model restrictive while our method adapts to the threshold by maximizing a simple reward function that maximizes the confidence while keeping the computational cost low. Also, note that if we test our algorithm on the dataset say IMDb with movie review classification, we train the backbone on SST-2 a sentiment classification dataset. Hence if we use the budgeting methods then we need a validation set to learn the $q$ values, which then makes the method offline which is not the case we considered.
>
> To the best of our knowledge, our proposed approach constitutes a zero-shot adaptation given the test data has a similar kind of task, uniquely tailored to accommodate distribution changes in the test dataset, a facet we believe has not been previously addressed in the existing literature.
>
> Also, I would like to refer to this paper "Adaptive Inference through Early-Exit Networks: Design, Challenges and Directions" which suggests that post-training these exiting policies need to be adapted to facilitate efficient deployment to use-cases with varying
> requirements and devices with different computational capabilities
>
> Q3:  In Tab. 2, the proposed method is compared with other baselines at a fixed computational cost. However, the main advantage of dynamic early exiting is one can adjust......
>
>
> A3: I guess by computational cost you mean the threshold for exiting For all the reported baselines, we do not change any of the parameters and directly use their codebases to evaluate the results. All the previous baselines in our paper use fixed threshold values for different layers of NLP tasks, we do not make any changes to their original ideas. Since our research was focused on NLP tasks, we compared it with all the works on NLP tasks and a very recent image captioning early exit model named "You need multiple exiting".
>
>
> Q2: It is recommended that........
>
> A2: We did not find these references in the existing literature of very recent NLP as well as Image captioning early exiting models. We will surely add them in the final version of the paper.

---

### Official Review · Reviewer_P3Sw · 2023-10-29

**Soundness:** 1 poor
**Presentation:** 2 fair
**Contribution:** 3 good
**Rating:** 5
**Confidence:** 4

**Summary:**

This paper proposes an online algorithm based on multi-armed bandits for adjusting the confidence threshold of BERT model with early exit gates. The objective is to reduce the model latency while maintaining high accuracy. The authors build on the existing rich literature of adding early exit classifiers on top of intermediate layers. Here, they focus on the challenge of selecting a good confidence threshold for deciding for each exit gate if to "exit" or not. Specifically the authors assume a domain shift of the test data and propose an online algorithm for adjusting the threshold according to the observed data.

A multi-armed bandit online algorithm is proposed for updating the exit threshold. The reward is designed as the difference in confidence between the last layer and exit layer subtracted by the increased cost, for instances that didn't exit. To normalize the two measures to a similar range, the cost $o$ is some value in [0,1].

First, an algorithm for single exit is described, then extension to multiple exits is presented. The experimental setting is focusing on OOD evaluation and examines the number of transformer layers computed vs. accuracy: training on one classification task and evaluating on another related dataset with similar classification task (repeated for 5 pairs).

**Strengths:**

Focusing on online adjustment of the early exit threshold is novel and interesting. The proposed method is based on multi-armed bandit and provides  an upper bound on the regret. Detailed algorithms are provided and experiments on 5 classification NLP datasets.

**Weaknesses:**

I value the novelty of the method and find it interesting. However, I see several weaknesses in the current paper:

1. While the proposed method is presented as general, the applicability beyond a single exit layer significantly increases the complexity and the solution space, possibly leading to long exploration stage before converging (the regret bound is only in expectation).
2. Also, many of the hyper-parameters feel pretty specific to the examined setting and justified in the paper with hand-wavy statements (e.g. "strategically positioned", "due to overthinking similar to overfitting during training" etc.), or with references to the appendix that don't fully explain them. This limits the generalizability of the solution.
3. The value of the cost parameter $o$ that is given to the end-user as a handle for controlling the desired cost is a uninterpretable value between [0,1]. Therefore, at the end of the day it feels like the user will still need to have some further calibration for tuning the value of $o$ to match whatever practical cost they can afford in their own measure and units.
4. While I see novelty and value in online adjustments of the threshold. The unsupervised novelty is less clear: see for example [1, 2, 3]. [1] and [3] seem to work with unlabeled data, and [3] seem to focus on threshold calibration which might be good to compare against.
5.  The experiments feel a bit underwhelming and unclear:
* The evaluation metric only measures the number of transformers layers and doesn't take into account any potential overhead of the exits and the calibration (and the use of "Time" as the column heading) is confusing.
* Since the method focuses on online setup, it would be interesting to see the patterns over time.
* The baselines model are not described well (for example, unclear what is the difference between ElasticBERT and DeeBERT).
* It is unclear how come the UBERT models could be better than the baselines in both accuracy and cost? If the backbone model is identical and roughly monotonic (as assumed throughout the paper), then the threshold should only control the tradeoff between the two but cannot improve on both?...

[1] https://aclanthology.org/2020.acl-main.537/

[2] https://aclanthology.org/2020.acl-main.593/

[3] https://aclanthology.org/2021.emnlp-main.406/

**Questions:**

see points in weakness section above. Also:
1. In eq.1 : are $C_p$ and $C_l$ always computed by max over softmax? the argmax can be different between the layer $p$ and $l$, making the use of the delta as reward less convincing.

---

> ### Author Response · Authors · 2023-11-21
> **Rebuttal**
>
> Thanks for the feedback!
>
> A1: Yes, for multiple exits the exploration period be large if we naively apply the classifical UCB. To address this issue, we have exploited the side observations to improve the convergence rate. And reduce exploration length. In Section 5, extension to multiple exits, we have discussed this. Also, in Proposition 5.1, we provide the probability of observing a particular arm without actually being played by the algorithm. Note that this probability increases with an increasing number of exits.
>
>
> A2: For positioning of early exits, we utilize some of the previous works referred to in the subsection 'exit selection'. We have referred to these papers where we mentioned this.
> Also, when we say "due to overthinking similar to overfitting", we provide the justification for why we are getting better accuracy even from the final layer. Overthinking is the process when a sample is correctly classified in some of the initial layers and in deeper layers, it is again misclassified.
>
>
> A3: We agree that the user has to provide a cost after calibrating it to match practical values. We have discussed clearly how to do this. For example, we have discussed about flops-based calibration. We believe that these calibrations are required to be input by the users as per their cost perception and there is no fixed way to define this.
>
>
> A4: Note that we are using pre-trained models. During the inference time, the data may not follow the same distribution as that used in the training and validation. This is indeed the practical case. In this inference stage, we may not have access to any ground truth label, However, we aim to find a threshold that is optimal. Here the optimality is with respect to the latent ground truth distribution. Hence we refer to our setup as unsupervised.
> Also note that in  [1, 2, 3], the validation data is assumed to follow the same distributions as that of training data. Whereas in our case, during the inference time, the data distributions of incoming samples may have deviated from that of the training data. For example, the ElasticBERT may have been tuned on SST-2, but inference is done on the IMDb and Yelp datasets.
>
>
> A5-(i) We note that we are focusing on NLP tasks using ElasticBERT backbone where each transformer layer is of the same size and complexity. Thus, the cost is directly proportional to the number of transformer layers used. In the evaluation metric, we have separately shown accuracy as well as the cost. The cost here corresponds to the overheads and can also be interpreted as time as each layer is of the same complexity.
>  Many previous works already treat it as time measurements to provide a fair comparison. Ex- "You need Multiple exiting" [tang et.al.].
>
>
> A5-(ii) For the online learning algorithms the patterns over time are observed by plotting the regret curves. We have already provided these details in Figures 8 and 9 in the Appendix which provide the cumulative regret over time.
>
>
> A5-(iii) The main difference between DeeBERT and ElasticBERT is: that DeeBERT uses a separate training where the BERT backbone is first fine-tuned the weights of the backbone are freeze and later exits are attached for further training while ElasticBERT does a joint training. We have detailed the setup of ElasticBERT in Appendix C. We will provide more details about the difference in the updated version.
>
> A5-(iv) The main advantage that UBERT has: is its adaptation of the threshold values. However, all the other baselines use a fixed threshold across all the exits so in those cases which were learned from the validation dataset, which was specific for that dataset. But during test time, the distribution of the sample could deviate from that of the training distribution which in turn changes the confidence distribution. Note that our finetuning and testing data are different.
>
> Also, fixed thresholds make the model more restrictive. We explained this in the paper in the results section "Previous methods like DeeBERT, ElasticBERT, PABEE, and MuE relied on fixed thresholds across layers and domains, resulting in either early exits from initial layers, leading to accuracy loss, or processing until the final layer, incurring higher inference costs.
>
> A6: Yes, Cp and CL are always computed by max over the softmax.
> Yes, the arg max can be different for different layers however if the given layer gives a misclassification then the confidence output of that layer will be very small. It is very rare to have a misclassification with very high confidence. Lower confidence will not let the sample exit the backbone and the sample will be processed to the next layer. Even if somehow the sample exits the backbone at the layer with misclassification then its reward will take a hit as lower confidence will in turn lower the reward value for that layer hence for future samples a higher threshold value will be chosen for that layer.

---

### Official Review · Reviewer_sWQ2 · 2023-10-31

**Soundness:** 2 fair
**Presentation:** 2 fair
**Contribution:** 2 fair
**Rating:** 5
**Confidence:** 1

**Summary:**

Inference latency is a key issue in any pre-trained large language models like BERT. Typically, side branches are attached at the intermediate layers with provision of early exit to minimize the inference time. This paper proposes an online learning algorithm, dubbed as, "UBERT" to decide if a sample can exit early through an intermediate branch.

**Strengths:**

1. Paper is well-written and the problem setup is mostly clear.

**Weaknesses:**

I am not an expert in this domain. However, I have few concerns.
1. Is it necessary to formulate the problem as multi-armed bandit setup? As RL usually resource hungry algorithms and they can take huge time to optimize.

**Questions:**

Refer to weaknesses section.

---

> ### Author Response · Authors · 2023-11-21
> **Rebuttal**
>
> Thanks!
>
> There are multiple ways in which the thresholds could be modeled e.g. using the validation dataset (in DeeBERT, BranchyNet, ElasticBERT, etc.), distributing the samples to each exit based on the budget available (MSDNet etc.). However, when it comes to adapting to the distribution changes i.e. when the backbone was fine-tuned on a training dataset and thresholds were learned on the validation dataset (with the same distribution) but if the test dataset has a different distribution than the training and validation dataset. We proposed an algorithm that when the latent data distribution of the incoming samples changes we can get good performance by just adjusting the thresholds without any further retraining.
>
> In our case, we are reducing the complete fine-tuning and validation task by introducing the RL algorithm.

---

### Official Review · Reviewer_nNLW · 2023-11-02

**Soundness:** 2 fair
**Presentation:** 2 fair
**Contribution:** 3 good
**Rating:** 3
**Confidence:** 4

**Summary:**

This work presents an early exit method for BERT. The authors use MAB to adaptively find threshold value in an online manner. The evaluation is conducted on 5 classification tasks.

**Strengths:**

1. The ablation seems good.
2. The concept of finding adaptive threshold seems interesting.
3. The proposed method outperforms the compared baselines.

**Weaknesses:**

1. Evaluation is restricted.
2. The authors claim are often missing supporting literature or validating experiments.
3. Authors exaggerates the efficacy of their method, even though they cannot also resolve the problem.
4. The paper is missing the essential information, which prevents the paper to stand alone.
5. Missing details.

**Questions:**

1-1 Considering recent SOTA NLP models/LLMs, BERT variants are quite older and their size is much smaller, which can already run smoothly with restricted resources under the current HW. Therefore, the only evaluation with BERT-variant in this work makes me doubt about the motivation of this work. The authors should've considered larger language models and showed the generalizability of the proposed work. If the scope is only limited to BERT, its applicability/practicability is questionable as BERT is barely used in real applications.

1-2 One of closely related work is F-PABEE that outperforms all existing methods in the literature. I highly recommend to add its result for comparison and analysis.

1-3 Other previous papers like PABEE and F-PABEE, CoLA MNLI MRPC QNLI QQP RTE SST-2 (STS-B) are standardized for comparison. However, authors does not follow.

2-1 "Even though it is anticipated that the final layer of the NN can have better accuracy than the intermediate layer": any support literature or experiments? This is the fundamental assumption, which is not validated throughout the paper.

2-2 "The threshold is often determined using a labeled dataset during training and serves as a crucial reference point for decision-making during inference.": in what cases or papers?

2-3 "The optimal threshold value depends on the distribution of confidence levels at the attached exit, which can vary depending on the data distribution. ": any proofs?

2-4 " UEE-UCB Hanawal et al. (2022) leverage the MAB framework to learn the optimal exit in EENNs": How is their use of MAB different from this work?

2-5 In Sec.2 in lines starting "LEE Ju et al. (2021b), DEE...", these works seem quite close to the proposed method. It would be recommended to have details comparison against the proposed work.

3-1 As authors noted, using a fixed threshold may yield suboptimal results. However, the proposed method finds the threshold based on the observations of previous samples, which cannot be also free from the same issue. So it seems "Consequently, UBERT sets itself apart" is not an appropriate claim.

3-2 The term online learning is quite confusing in this work. As in Sec. 6, the pretrained model is finetuned and this finetuned model is used to adaptively find threshold in an online manner. The adaptive finding is of course online, but this term (online learning/algorithm) is exaggerated and providing confusion.

3-3 The authors keep using the term "optimal threshold" throughout the paper. However, it is optimal only if the given specific setting in Algorithm 1 and 2 is used. With naive changing the cost such as adding a value or scaling, it varies. It is hard to conclude that the proposed method optimally trade-off between latency and accuracy. Is there a curve, for example, UBERT-2 shows best accuracy with -59.5 time while the accuracy reduces with -58 or -60 time? If not, the use of this term seems not proper in this context.

4-1 The detailed description of ElasticBERT and MAB is not provided.

5-1 "Though confidence and latency are in different units, we add them after using a conversion factor.": I cannot find details of this process in the paper.

5-2 What should I do if I want to improve the latency while sacrificing the performance or vice versa? The new model should be trained again? If so, although the authors adaptively find the threshold with reward, it is hard for me think it as a benefit compared to other exiting methods. Other work simply change the number and run the model to adjust latency-performance.

---

> ### Author Response · Authors · 2023-11-21
> **Rebuttal**
>
> Thanks for the reviews!
>
> A1: For simple tasks such as text classification, we believe that even the BERT model is overparameterized. Also, not only having resources is enough but also the efficient exploitation of these resources is important. Another reason to use ElasticBERT backbone is that it is a pre-trained model with a joint training strategy hence very minimal fine-tuning is required to create the early exit model for specific tasks.
>
> However, the scope is not limited to the BERT model but in this setup, we consider only text classification models and BERT still performs comparable to the SOTA NLP models while maintaining a smaller size. Hence ElasticBERT becomes a good testbed for our algorithm.
>
> A2: Since it is very recently published, we could not add this as our baseline for comparison as well as reference, however, we added older version of this PABEE as one of our baselines.
> We will surely compare our method with F-PABEE.
>
> A3: We already compared our work with PABEE, and also we used the datasets such as MNLI, MRPC, QNLI, QQP, RTE, and SST-2. However, we used some of them as pre-training datasets. For instance, from Table 1, we use the MRPC for fine-tuning the backbone and then we test it on the QQP dataset where not a single ground-truth is used from QQP. In table 1, we use all the datasets from GLUE and ELUE tasks.
>
> A4: In the appendix, we added Figure 2 which provides the accuracy of different layers for two datasets which is seen to be increasing, this suggested that the accuracy increases as we go deeper into the backbone and start saturating as well. This was the case with other datasets as well. Due to space constraints, we could not add this figure to the main body.
>
> Please note that here we are talking about the average accuracy of the complete dataset. However, for a subset of the dataset, things might not be the same.
>
> A5: All the referred papers in our work except AdaEE, use a threshold value that is either learned on a labeled validation set or it is fixed to maintain a good speed-up ratio on the validation set. The most relevant are: DeeBERT, ElasticBERT, "You need multiple exiting", PABEE, etc.
>
> A6: Yes, we experimentally validate this claim, IMDb and Yelp are two datasets that are similar to each other, one has movie review classification and the other has reviews from hotels, restaurants, etc. Hence these two datasets have different distributions (or domains) but similar tasks. We provide the confidence values in Figure 3, which show that there is some difference between these values and the confidence gain analysis (figure 4, 5) also states that the gain in confidence also changes with change in distribution.
>
> A7: Their objective is different from ours, they are trying to find the optimal layer (layer up to which DNN should be deployed on the edge device so that a particular text classification task is efficiently completed.) that should be incorporated on the edge device. After choosing the split, they deploy the DNN till the splitting point on the edge device to perform inference on edge only. In contrast, our work utilizes the MAB setup, to adaptively adjust the thresholds based on the task at hand and changing the distribution of the dataset.
>
> A8: We could not make a direct comparison with these methods as they are developed to find the best intermediate classifier in a service outage scenario such as network connectivity issues etc. In contrast, our work finds the best threshold given latency costs, however, they also use MABs to find the best intermediate classifier for a given service outage scenario.
>
> A9: By fixed threshold we mean similar thresholds across different layers, but in our method, every layer has a different threshold value that is adapted based on the confidence distribution at that exit point.
>
> By fixed thresholds, we also mean that the threshold values should be adapted based on the changing distribution during test time and not merely applying the same threshold that was learned at the training time. However, after convergence to the optimal threshold values given the cost structure, the threshold might be fixed in our setup as well but with different values for different distributions as well as different exit layers. It is not fixed across the changing distribution of datasets.
>
> A10: Since the data arrives in an online fashion, and for an incoming sample, our algorithm on the fly decides the threshold values hence the threshold is being learned when data arrives in an online fashion, hence, we call our method an online learning algorithm. Our method updated the reward function given in Equation 1 after it received a sample, hence it also learns in an online fashion.

---

> > ### Author Response · Authors · 2023-11-21
> > **Answers to other questions**
> >
> > A11: Observe that the input to algorithms 1 and 2 are the latency costs and the exploration parameter, The exploration parameter decides if the user wants to explore more or exploit early, hence we fix it to 2. The only thing that our algorithm depends on is the input latency costs. You are right that the optimal threshold is only with respect to the values of input to the algorithm which should always be the case. Based on the cost structure given by the user, it will find the optimal thresholds to decide an early exit depending on the distribution of the dataset and the computational power of user's device.
> >
> > To make it clear, we only say optimal thresholds based on the given latency cost values. We have already stated that there is a accuracy-cost trade-off, hence if one of the metric improves other declines. Hence we are optimal in modelling this trade-off given the latency cost at the user's end.
> >
> > A12: The description of ElasticBERT is given in the Appendix C. Also, MAB framework is very classic framework to solve the online learning problem, hence, we do not provide details on that.
> >
> >
> > A13: The details are provided in the paper, we have a subsection named "latency cost" inside the experimental section, where we have provided the details of the how we scale the latency cost and how users can choose them.
> >
> > A14: If the user wants to improve the latency while sacrificing the accuracy then increase the latency cost values to higher values that are input to the algorithm. Higher latency costs will increase the negative impact of processing the sample deeper in the backbone. If a sample is processed till deeper layers due to higher threshold values chosen for initial layers, then the reward will take a hit and then the chosen action values will lower its reward and will not be explored much. In our case also, it is easy